# Hematopoietic Stem Cell Gene-Addition/Editing Therapy in Sickle Cell Disease

**DOI:** 10.3390/cells11111843

**Published:** 2022-06-04

**Authors:** Paula Germino-Watnick, Malikiya Hinds, Anh Le, Rebecca Chu, Xiong Liu, Naoya Uchida

**Affiliations:** 1Cellular and Molecular Therapeutics Branch, National Heart Lung and Blood Institutes (NHLBI)/National Institute of Diabetes and Digestive and Kidney Diseases (NIDDK), National Institutes of Health (NIH), Bethesda, MD 20892, USA; malikiya.hinds@nih.gov (M.H.); anh.le2@nih.gov (A.L.); rebecca.chu@nih.gov (R.C.); liux@nhlbi.nih.gov (X.L.); 2Division of Molecular and Medical Genetics, Center for Gene and Cell Therapy, The Institute of Medical Science, The University of Tokyo, Tokyo 108-8639, Japan

**Keywords:** hematopoietic stem cells, transplantation, gene therapy, gene editing, sickle cell disease, in vivo gene therapy

## Abstract

Autologous hematopoietic stem cell (HSC)-targeted gene therapy provides a one-time cure for various genetic diseases including sickle cell disease (SCD) and β-thalassemia. SCD is caused by a point mutation (20A > T) in the β-globin gene. Since SCD is the most common single-gene disorder, curing SCD is a primary goal in HSC gene therapy. β-thalassemia results from either the absence or the reduction of β-globin expression, and it can be cured using similar strategies. In HSC gene-addition therapy, patient CD34+ HSCs are genetically modified by adding a therapeutic β-globin gene with lentiviral transduction, followed by autologous transplantation. Alternatively, novel gene-editing therapies allow for the correction of the mutated β-globin gene, instead of addition. Furthermore, these diseases can be cured by γ-globin induction based on gene addition/editing in HSCs. In this review, we discuss HSC-targeted gene therapy in SCD with gene addition as well as gene editing.

## 1. Introduction

Sickle cell disease (SCD) was first reported by James Herrick in 1910 who described the presence of ‘peculiar elongated and sickle-shaped red blood corpuscles’ [1]. Currently, millions of people around the world (~100,000 in the USA) are suffering from SCD, and approximately 300,000 babies (~3000 in the USA) are born each year with SCD [2,3]. SCD is the most prevalent inherited single-gene disorder, caused by a point mutation of Glu6 to Val through a single nucleotide conversion of A-to-T at position 20 in the *HBB* gene that encodes the β-globin chain of hemoglobin (Hb). In erythrocytes, two α-globin chains and two β-globin chains associate into a tetramer called adult hemoglobin (HbA), which binds oxygen with high affinity and delivers it to all tissues. For SCD patients, under hypoxic conditions, the mutant sickle hemoglobin (HbS) polymerizes into numerous fibers, ~21 nm in diameter and up to several μm in length. As the HbS bundles together in parallel, red blood cells are distorted, forming the characteristic ‘sickle’ shape. This results in hemolytic anemia, vaso-occlusive crises (VOC), systemic inflammation, and multi-organ damage [4]. SCD reduces a patient’s life expectancy by several decades [5]. β-thalassemia is a similar genetic disease, caused by the absence or reduction of β-globin expression due to various mutations and deletions [6]. β-thalassemia leads to severe anemia but not VOC, and it can be cured by similar treatment as SCD. In HSC gene therapy, the same therapeutic globin vectors are applicable for both β-thalassemia and SCD. However, β-thalassemia is thought to be curative at lower levels of lentiviral gene marking than SCD, and thus, it was the subject of earlier gene therapy trials. The therapies developed and optimized using β-thalassemia are currently applied in SCD. In this review, we will predominantly discuss SCD but will touch on β-thalassemia as well.

Current treatment options for SCD are supportive care (medications and blood transfusions) and allogeneic transplantation of hematopoietic stem cells (HSC) from a healthy donor. Four drugs (hydroxyurea, glutamine, voxelotor, and crizanlizumab) have been approved by the US Food and Drug Administration (FDA) for the treatment of SCD [7,8]. Their therapeutic mechanisms and efficacies continue to be thoroughly investigated [9]. Clinical data reveal that these drugs work best for patients with mild SCD. For patients with severe SCD, the best treatment option is allogeneic HSC transplantation. However, only ~10% of patients with SCD have a histo-compatible sibling donor. Additionally, the risk of morbidity and mortality still exists with HSC transplantation due to conditioning as well as suboptimal donor matching. However, preliminary genetic engineering studies reveal the promise of these newer therapies [10,11,12,13,14,15,16]. As such, gene therapy followed by autologous HSC transplantation has been vigorously pursued and holds promise for curing SCD.

Recently, HSC gene therapy has been employed in the treatment of various hereditary genetic diseases, including primary immunodeficiencies, hemoglobinopathies, inherited bone marrow failure, and metabolic diseases [17,18,19,20]. While it is currently being applied in SCD, achieving efficient gene marking at the HSC level and robust globin expression in erythroid cells remains a formidable obstacle [21]. Despite these challenges, tremendous advances in HSC gene therapy have been seen in the past decade, which is paving the way for the clinical use of ex vivo gene therapy. In brief, the HSC gene therapy process for SCD involves harvesting bone marrow HSCs, selecting and genetically modifying the CD34+ cells, and ultimately reinfusing the engineered cells back into the patient (Figure 1).

Multiple strategies have been developed for the genetic manipulation of CD34+ cells. First, with the advent of newer editing technologies, the mutated β-globin gene itself can be corrected with HSC-targeted gene-correction therapy [22,23]. While healthy HbA expression is the main goal of gene therapy for individuals with SCD or β-thalassemia, HbF induction also has ameliorative effects in these patients. HbF is composed of two α-globins and two γ-globins, lacking the mutated β-globin. Additionally, it has anti-sickling effects as well. Targeting γ-globin expression through its regulatory elements such as *BCL11A,* is another approach for HSC editing. The gene-addition/editing strategies we will cover can be loosely described as (1) lentiviral vector (LV)-mediated insertion of a therapeutic β-globin or γ-globin gene, (2) gene editing to repair the mutant SCD codon, (3) LV-mediated *BCL11A* silencing (a γ-globin repressor), and (4) targeted disruption of HbF regulatory elements. In this review, we describe each strategy, update the recent progress and cases of clinical trial studies, and discuss the perspective of HSC gene therapy for SCD and β-thalassemia.

## 2. HSC-Targeted Gene Therapy with Lentiviral Gene Addition in SCD

In HSC-targeted gene therapy, patient HSCs are genetically corrected by β-globin gene addition, followed by autologous transplantation, allowing for a one-time cure of SCD. LVs are a popular genetic viral delivery method. They are lentiviruses composed of a single-stranded RNA genome capable of transducing both dividing and quiescent cells. Since they integrate into the host genome, they provide long-term and stable gene expression. However, insertional mutagenesis remains a risk. Additionally, achieving long-term engraftment of gene-modified cells post-transplant can be challenging as culture conditions must adhere to certain guidelines.

### 2.1. HSC Collection from Patients with SCD

HSCs are multipotent cells found predominantly in the bone marrow that can differentiate into any type of blood cell. They express CD34 and constitute blood for a lifetime. While technologies to harvest HSCs have improved, some difficulties remain. HSCs can be extracted from either the bone marrow directly or from the peripheral blood. Steady-state bone marrow CD34+ cells can be harvested from SCD patients [24], but they often contain significant amounts of erythroid progenitors, reducing the efficacy of lentiviral gene addition [25]. Furthermore, the one-time yields of CD34+ cells harvested from bone marrow are insufficient for gene therapy. Therefore, a few cycles of bone marrow harvests are needed per patient. As this process is invasive, physicians have turned to peripheral blood-derived HSCs as an alternative. HSCs can be mobilized to the peripheral blood from the bone marrow via plerixafor, a *CXCR4* antagonist [26]. Then, they are harvested by leukapheresis and the CD34+ cell fraction can be isolated by magnetic beads. Granulocyte colony-stimulating factor (G-CSF) is commonly used for HSC mobilization from healthy donors and patients with hematologic malignancies. However, G-CSF is contraindicated in SCD patients as it induces VOCs, multi-organ failure, and even death. The plerixafor/G-CSF combination increased CD34+ cell collection yields ~2-fold, compared with G-CSF alone [27,28]. However, this combination is also contraindicated in SCD.

Interestingly, factors such as disease severity, hydroxyurea treatment, and age affect HSC collection yields [29]. CD34+ cell yield is negatively correlated with markers of disease severity and hospitalization frequency. Additionally, both hydroxyurea treatment and an increase in age reduce HSC collection rates. Specifically, one study showed that HSC yields in individuals around 30 years of age declined to around <2 × 10^6^/kg after 2 cycles of mobilization/apheresis [29]. This reduction may be due to cell damage caused by chronic inflammation in SCD and years of HSC niche disruption. As a result, patients should have HSC mobilization/collection before 30 years of age and end hydroxyurea treatment 20–30 days before HSC collection [29].

Recently, a new multivariate model was developed to estimate the total CD34+ cell yield on the first day of apheresis [27]. This model allows practitioners to predict poor mobilization conditions with 85.7% accuracy [27]. In addition to plerixafor, other novel, unapproved mobilization agents, such as POL6326 [30,31], BKT140 [32], parathormone, and BIO5192 [33] are being tested. Nonetheless, until these agents are approved and optimized, plerixafor followed by apheresis remains the safest harvesting strategy.

### 2.2. HSC Culture Conditions for Gene Addition and Engraftment

Ex vivo culture with cytokine stimulation is required for lentiviral transduction in CD34+ HSCs. However, overstimulation in CD34+ cell culture reduces their engraftment ability. One-day pre-stimulation with serum-free culture media including cytokines (stem cell factor (SCF), FMS-like tyrosine kinase 3 ligand (FLT3L), and thrombopoietin (TPO), 100 ng/mL each) followed by one-day transduction with an LV allows for robust engraftment and efficient *EGFP* gene marking in CD34+ cells in xenograft mice [34,35]. A lower concentration of SCF enhances the engraftment of CD34+ cells, and serum albumin can be replaced with polyvinyl alcohol [36]. Long-term culture on fibronectin-coated plates allows for the engraftment of CD34+ cells, but does not enhance engraftment [35]. High-density culture with adjuvants such as dimethyl-prostaglandin E2 (PGE2) and amphiphilic drug-delivery poloxamers can improve transduction efficiency ~10-fold in human CD34+ cells in vitro [37]. Overall, cytokine stimulation is required for lentiviral transduction in ex vivo CD34+ cell culture. However, minimal stimulation and short-term culture are preferred to maintain the balance between efficient gene addition and robust engraftment of CD34+ cells.

A meticulously designed vector is also critical for high-level transduction and β-globin expression at the therapeutic level. Therapeutic genes, such as wild-type β-globin, β^T87Q^-globin (containing an anti-sickling mutation), or γ-globin are usually inserted into a self-inactivating (SIN) LV. Additionally, a β-globin promoter and the locus control region (LCR) are inserted to control transgene expression [21,38,39]. As the 2nd intron of the β-globin gene is critical to its expression, traditional vectors carry β-globin cassettes in the opposite direction of the vector genome to prevent transcriptional excision. Since this reversed construct reduces vector titers and transduction efficiency in CD34+ cells, a forward-oriented β-globin vector was recently developed. It addressed these issues and produced robust β-globin expression in erythroid cells [21,38].

### 2.3. Myeloablative Conditioning for Engraftment of Gene-Modified Cells

Efficient engraftment of transplanted HSCs requires conditioning generally involving total body irradiation and/or high-dose chemotherapy before HSC infusion [39,40,41]. Allogenic HSC transplantation conditioning can be divided into two categories: (1) myeloablation to open the HSC niche space in host bone marrow, improving donor cell engraftment, and (2) immunosuppression to prevent immunological rejection of graft cells by the host. Immunosuppression is not necessary for HSC gene therapy, since gene-modified cells are derived from autologous CD34+ cells. However, myelosuppression is still essential to opening the bone marrow niche [42,43]. Therefore, high-dose busulfan (myeloablative but not immunosuppressive) is utilized for conditioning in gene therapy to deplete host HSCs. Short- and long-term toxicity in hematopoietic and non-hematopoietic tissues poses a challenge in myeloablative conditioning for autologous HSC gene therapy. To mitigate the toxicity of conditioning, the dosage is adjusted based on therapeutic levels of lentiviral gene marking necessary for disease correction [44].

Reduced-intensity conditioning (RIC), which reduces chemotherapy-associated toxicities, is also a viable alternative [14]. However, it can lead to lower levels of lentiviral gene marking and inefficient engraftment due to only partial extermination of host HSCs [45]. Recently, scientists have harnessed monoclonal antibodies to selectively target HSCs to achieve stable engraftment and reduce toxicities normally associated with myeloablation. Specifically, CD117 (*c-KIT*) is expressed in HSCs and hematopoietic progenitors, but not lymphocytes. Therefore, it serves as an optimal target for myeloablative conditioning without immunosuppression. One study showed that CD117-antibody-drug conjugates (ADCs) were able to effectively deplete human HSCs in xenograft mice, while preserving immunity and allowing for efficient engraftment of gene-modified cells in non-human primate models [46,47,48]. Additionally, the unconjugated CD117 antibody with 5-azacytidine was also able to deplete HSCs in a mouse model [49].

### 2.4. Potential Insertional Mutagenesis in Gene-Addition Therapy

A limitation of retrovirus-based vectors, such as γ-retroviral vectors and LVs, is the potential for integration into transcriptionally active regions of the host’s genome. Specifically, γ-retroviral vectors tend to integrate around transcription start sites near promoters. Some preliminary HSC gene therapy trials for primary immunodeficiencies that used γ-retroviral vectors resulted in hematologic malignancies due to insertional mutagenesis [50]. For example, γ-retroviral insertion into an oncogene (*LMO2)*, induced T-cell leukemia development in an early gene therapy trial in X-linked severe combined immunodeficiency (X-SCID) [51,52,53]. Therefore, a safer LV system has been developed from human immunodeficiency virus type 1 (HIV-1), in which the promoter and enhancer regions of the long terminal repeats (LTRs) have been deleted to create SIN-LVs [54].

Currently, there have been two reported cases of myeloid malignancies—myelodysplastic syndrome (MDS) and acute myeloid leukemia (AML)—in two LV-based gene therapy clinical trials for SCD (ClinicalTrials.gov: NCT02140554, NCT04293185). While the bluebird bio trials were initially halted, further analysis revealed these malignancies did not develop from LV insertional mutagenesis. The first case of MDS, which eventually transformed into AML, was reported in a patient 3 years post-gene therapy [55]. It was determined that their AML developed from busulfan conditioning due to the absence of vector integration in the leukemic cells. However, in the second case of AML, lentiviral integration was detected in leukemic cells. Nonetheless, further analysis revealed that the LV was an unlikely cause of leukemia development, since the integration site was found at the 4th intron of the *VAMP4* gene (with no known role in cell proliferation or oncogenesis). Additionally, the gene expression profile was not changed around the *VAMP4* gene [56]. Furthermore, SCD patients are more likely to develop MDS/AML along with chromosomal abnormalities and oncogene mutations, which were also observed in the two cases of leukemia post-gene therapy [57,58]. Additionally, the leukemic clones found in the MDS/AML patients post-allogenic HSC transplantation were identified in the pretransplant samples at low levels [59]. Taken together, the data suggest that pre-existing leukemic clones related to SCD were predominantly selected due to busulfan conditioning and/or the low clonal variance of HSCs post-gene therapy, resulting in MDS/AML development.

### 2.5. Clinical Trials of Gene-Addition Therapy in SCD and β-Thalassemia

While the first gene therapy trials were initially employed in the treatment of X-SCID, lentiviral gene addition is now being used for hemoglobinopathies (Table 1). The first gene therapy patient with transfusion-dependent β-thalassemia (TDT) was part of the LG001 trial in France in 2006. After undergoing bone marrow harvest and busulfan conditioning, the patient was transduced with the HPV569 LV, a SIN-LV encoding the β^T87Q^-globin variant flanked by two LTRs, each containing two copies of the core 250 bp chicken hypersensitivity site 4 (HS4) insulator. They became transfusion-independent after one year with Hb levels remaining stable at ~8.5 g/L for more than 8 years with the therapeutic Hb accounting for >30% of the total [60]. Although this trial and the use of HPV569 were ultimately terminated due to vector insertion inside the *HMGA2* oncogene, it played a critical role in informing future studies. First, it revealed the difficulty in obtaining sufficient bone marrow harvest, transduction, and engraftment leading to the employment of mobilization agents such as G-CSF and plerixafor. Secondly, the core HS4 insulator was determined to cause greater genomic instability and lower titers [61]. Its removal from the U3 region (an original HIV-1-derived promoter) plus the incorporation of the cytomegalovirus promoter to increase vector transcription led to the creation of a new LV, BB305 [60,62]. Thirdly, the *HMGA2* integration resulted in clonal dominance without leukemia development. This is probably due to the low clonal variance of gene-modified HSCs, but not insertional mutagenesis. Thus, both robust engraftment of gene-modified cells and high-level lentiviral gene marking might be important to allow for the generation of polyclonal hematopoiesis as well as the prevention of clonal dominance.

The next generation of clinical trials sought to build upon these insights. In 2013, bluebird bio sponsored two studies that employed the BB305 LV to insert the anti-sickling β^T87Q^-globin variant into patients with SCD and TDT (HGB-204 and HGB-205, respectively). The studies used a combination of G-CSF and plerixafor for mobilization. Of the 22 individuals with TDT treated, 12/13 with milder TDT and 3/9 with more severe TDT became transfusion independent while the other 6/9 had reduced transfusion requirements (NCT02633943) [63,64]. Following the HGB-204 and HGB-205 studies, two Phase 3 studies were initiated: one to further assess the clinical benefits of BB305 in less severe β-thalassemia (HGB-207) and the other to study the efficacy of G-CSF/plerixafor mobilization in individuals with both β-thalassemia minor and major genotypes (HGB-212). HGB-207 reported vector copy numbers (VCNs) greater than those in HGB-204 and HGB-205 studies (1.9–5.6 vs. 0.3–2.1). In the HGB-207 and HGB-212 studies, 92.30% and 77.78% of patients <18 years of age have stopped transfusions for ≥6 months, respectively [65]. Though bluebird bio was temporarily forced to suspend these trials in 2021 due to a serious adverse event (SAE) of AML, they have recently restarted them as they concluded that vector integration was unlikely to play a role in the patient’s development of AML [66].

In addition to BB305, other trials employing the TNS9.3.55 and GLOBE vectors to insert β-globin into TDT patients have been underway as well. The first clinical trial in the USA in 2012, employed the TNS9.3.55 vector, which included the wild-type β-globin gene, larger LCR fragments, and a longer β-globin promoter sequence. Additionally, G-CSF was used for CD34+ mobilization rather than bone marrow harvest and busulfan was used for RIC (8 mg/kg busulfan). While only four patients were enrolled, the study reported low VCNs (0.09–0.15) and patients did not attain transfusion independence. The degree to which their symptoms were abrogated was dependent upon VCN and the severity of the initial disease [45,67]. In Milan, nine patients were treated with the GLOBE vector, which incorporates only HS2 and HS3, omitting the HS4 element of the β-globin LCR. The removal of HS4 increases the viral titer with similar gene expression. Patients were conditioned with treosulfan and thiotepa instead of busulfan to reduce toxicity, then the gene-modified cells were delivered intraosseously. The four adult patients treated experienced a significant reduction in transfusion requirements, while 4/5 children achieved transfusion independence [68,69]. Two new studies have begun in China employing two new LVs. One is a Phase 1 study to evaluate the safety and efficacy of the BD211 drug product in non-β0/β0 TDT major patients (NCT05015920). The other is a Phase 1 trial that is not yet recruiting, but will be studying the ability of the LentiHBBT87Q system to restore the β^T87Q^-globin expression in pediatric TDT major patients (NCT04592458).

The first patients with SCD to participate in a gene therapy trial were part of the bluebird bio HGB-205 study receiving modified CD34+ cells transduced with the BB305 LV carrying the anti-sickling β^T87Q^-globin gene (NCT02151526). Of the three patients enrolled, two patients had sustained improvement in their disease, achieving transfusion independence, and one saw transfusion reductions. The anti-sickling Hb (non-HbS) achieved stable expression with overall non-HbS accounting for 40–52% of total Hb in these two patients, similar to sickle cell trait [62]. Following the success of this trial, bluebird bio initiated another Phase 1/2 trial (HGB-206) solely for SCD patients and a Phase 3 trial (HGB-210) (NCT02140554, NCT04293185). In the HGB-206 study, seven patients were initially enrolled under the Group A protocol. However, due to low drug product VCN (transduced CD34+ cells) and insufficient HbA^T87Q^ expression, two additional study groups were created (B and C). Only two patients were enrolled under Group B, which included pre-harvest red blood cell transfusions, higher target busulfan levels, and refined drug product manufacturing. Group C patients were treated under a revised protocol that specified drug products made from plerixafor-mobilized HSCs (NCT02140554). Five out of seven Group A participants and both Group B participants did not require regular red blood cell transfusions post-gene therapy (Group A average: drug product VCN 0.6, HbA^T87Q^ 0.5 g/dL) [70]. Nonetheless, Group C (*n* = 35) updates, last reported in December 2021, produced better results (Group C: drug product VCN 3.7, HbA^T87Q^ 5.2 g/dL). Additionally, Group C patients exhibited higher pan-cellular expression, poly-clonality, and elimination of severe VOC events [71]. HGB-210 will be a Phase 3 continuation of HGB-206 and will also evaluate LentiGlobin BB3305, though no results are available to date.

Other Phase 1/2 trials are testing LV-based clinical protocols. Aruvant Sciences is using their ARU-1801 LV carrying the γ-globin gene combined with a RIC regimen (single dose of melphalan) (NCT02186418). As of 2020, Patients 1 and 2 had around 20% inserted HbF with 30% total non-HbS 30 months post-transplant. Patient 3 had 24% inserted HbF and 38% total non-HbS 6 months post-transplant. Meaningful VOC improvement was seen in all three patients [72]. The DREPAGLOBE trial using the GLOBE1 LV expressing the β^AS3^-globin gene (G16D, E22A, and T87Q) reported variable efficacy (NCT03964792). Though 2/3 patients received treatment benefits, therapeutic amelioration correlated with VCN, reflective of engraftment capability. Patient 3 remained transfusion dependent due to low VCN and <3.0% therapeutic Hb [73]. A different study out of the University of California, Los Angeles, is using the β^AS3^-FB carrying the β^AS3^-globin gene and including an enhancer-blocking insulator in the 3′LTR (the footprint II (FII) of chicken HS4 insulator and human T-cell receptor blocking element alpha/delta 1 (BEAD-1) insulator). Results have yet to be reported (NCT02247843). Boston’s Children’s Hospital took a different approach, transducing CD34+ HSCs with an LV (BCH-BB694) containing a microRNA-adapted short-hairpin RNA (shmiR) targeting the *BCL11A* gene. As of 2020, six patients had achieved robust and stable HbF induction with 30.5% HbF of all Hb levels and 70.8% of HbF-positive red blood cells (F-cells) [74]. This trial was temporarily suspended in 2021 over potential mutational mutagenesis [66]. Finally, a Phase 1 pilot study using CSL200 and RIC melphalan conditioning was terminated due to unexpected delays (NCT04091737). CSL200 is a drug product transduced with the CAL-H LV encoding γ-globin^G16D^ and a short-hairpin RNA (sh-734). The γ-globin point mutation (G16D) leads to increased γ-globin production compared to endogenous γ-globin and sh-734 targets the *HPRT* gene increasing positive gene selection (NCT04091737).

## 3. HSC-Targeted Gene-Editing Therapy in SCD

While SCD gene-addition clinical trials utilizing LVs have been underway for the past 15 years, gene-editing trials have just taken off. Gene editing relies on endogenous cellular repair mechanisms, endonucleases, bacterial defense pathways, and crafty editing toolkit delivery systems. Though advancements have been made in the field allowing for the approval of such clinical trials, gene-editing techniques require further optimization as efficient editing and long-term engraftment remain elusive (Figure 2).

### 3.1. Endogenous DNA Repair Mechanisms

Gene-editing technologies rely on endogenous cellular DNA repair mechanisms. When double-strand breaks (DSBs) occur in the cell, non-homologous end joining (NHEJ) and homology-directed repair (HDR) are the two primary methods for correction. NHEJ is the process by which the cell ligates two random strands together, which can introduce insertions and deletions (indels). In contrast, HDR uses a homologous template (donor DNA), such as an unbroken sister chromosome or an artificial DNA strand, to synthesize a new copy and facilitate damage repair [75]. Compared to LV gene-addition therapy, the gene-correction approach has several advantages including avoiding insertional mutagenesis and allowing gene expression to be maintained by endogenous promoters and regulatory elements [52]. While NHEJ can occur during any phase of the cell cycle, HDR is restricted to the late S phase or G2 phase. Since indels created by NHEJ usually result in non-functional gene products, they are generally used for knockouts. For example, it is used to disrupt the binding sites for erythroid-specific transcriptional activators related to Hb switching (i.e., *BCL11A*). HDR is less error-prone and usually used for homologous recombination, such as replacement of the SCD mutation on the β-globin gene.

### 3.2. Engineered Endonucleases for Site-Specific DNA Break

Engineered endonucleases can recognize the SCD mutation in patient CD34+ HSCs, allowing for site-specific DNA cleavage. Preliminary studies utilizing endonucleases revealed low correction efficiencies, malignant transformations, and high rates of cell death [76]. Currently, they are incorporated via three important site-specific endonucleases: zinc finger nucleases (ZFNs), transcription activator-like effector nucleases (TALENs), and clustered regularly interspaced short palindromic repeats (CRISPR)/CRISPR-associated protein 9 (Cas9) [77]. These newer methods allow for the enhancement of HDR with minimal mutations/deletions, reduced off-target effects, and higher editing efficiencies.

Though ZFNs and TALENs are valuable tools, they are costly, time-consuming, labor-intensive, and require expertise in protein engineering to design specific nucleases. In contrast, the CRISPR/Cas9 gene-editing system has shown better correction efficiency as well as easy design for the target sites [78]. It is derived from the innate immune system of the *Streptococcus pyogenes* bacteria. CRISPR editing consists of an endonuclease (often Cas9) and a single guide RNA (sgRNA). The sgRNA includes a ∼20 nucleotide target sequence and an ∼80 nucleotide RNA scaffold [79]. To edit with CRISPR, there must be a protospacer-adjacent motif (PAM) downstream of the target sequence. For Cas9, this is usually a 2–4 nucleotide guanine-rich region [79]. Other Cas endonucleases can be used as well, such as Cas12, which has a thymine-rich PAM region, and Cas13, which is an RNase as opposed to a DNase.

However, CRISPR/Cas9 gene correction efficiency is still limited by off-target activity [80,81]. This disrupts normal gene function and genomic instability, which could lead to oncogenesis, a major concern in clinical studies. In addition, pre-existing antibodies to Cas9 were reported in a large-scale clinical study in the U.S. [82,83]. Though the relevance of these findings remains unknown, it ought to be considered as CRISPR’s application expands.

### 3.3. Gene-Editing Delivery Methods

CRISPR/Cas9 system delivery mechanisms also affect editing efficacy. Editing machinery can be delivered in three forms: (1) DNA, (2) RNA, or (3) ribonucleoprotein (RNP) complexes. Delivery systems can be classified as viral or non-viral (physical vs. chemical). Viral vector systems mainly include lentiviruses, adenoviruses (Ads), and adeno-associated viruses (AAVs). Non-viral physical systems include electroporation and microinjection, while non-viral chemical systems include lipid nanoparticles (LNPs) and gold nanoparticles (AuNPs) [84].

Electroporation is a form of transfection that creates temporary pores in the cell membrane via an electrical pulse. It can simply and reliably deliver CRISPR cargo (for DNA, RNA, or RNP) to various cell types. In most cases, electroporation-mediated delivery is used for ex vivo gene editing, such as ex vivo HSC gene-editing therapy. However, it is toxic and often results in low cell viability, and therefore, further optimization is preferred. Microinjection delivers editing machinery under a refined microscope using a 0.5–5.0 µm microneedle. It has ~100% efficiency, low toxicity, and can deliver all CRISPR cargo sizes. Nonetheless, it is only optimal when editing a small number of cells. LNPs and AuNPs are alternative delivery techniques. Nanoparticles can be used for in vitro and in vivo gene delivery, but have been mainly developed for in vivo gene editing.

As gene-addition therapy has gained traction, viral vector delivery systems have also received more attention and development. However, long-term transgene expression from viral vectors is not suitable for gene editing, as it can increase genotoxicity due to off-target effects. Therefore, the use of a viral vector system is limited in ex vivo gene editing. Nonetheless, short-term exposure is sufficient and safe. AAV vectors are predominantly non-integrating, with a small risk of random insertion. Unfortunately, they have a limited DNA payload (<4.7 kb), making it difficult to carry the Cas9 cDNA [85]. However, along with electroporation-based CRISPR/Cas9 delivery, AAV vectors can be used for larger donor DNA delivery.

### 3.4. Gene Correction of the SCD Mutation with Gene Editing

CRISPR has been shown to have higher editing efficiencies and is easier to use for target modifications than ZFNs and TALENS, making it the dominant editing technique for SCD. Using CRISPR/Cas9, scientists have achieved mutant *HBB* correction both ex vivo and in vivo. This technique combines a site-specific sgRNA, Cas9 endonuclease, and donor DNA flanked with homology arms encoding the correct β-globin sequence. First the sgRNA guides Cas9 to the desired editing location, Cas9 creates a DSB, and then using HDR, the cell will ligate in the corrected donor gene. This RNP complex is most commonly delivered via electroporation, which is restricted to ex vivo culture, or viral vectors and nanoparticles, which can be used both ex vivo and in vivo.

Several studies have been able to insert the corrected β-globin gene ex vivo with varying levels of efficiency. An earlier study attained over 18% gene modification in bone marrow-derived CD34+ cells from SCD patients [78]. AAV vector-based donor DNA delivery resulted in 90% targeted AAV insertion in the β-globin gene in CD34+ cells and 4–30% engraftment in xenograft mouse transplantation [23]. A different study was able to achieve 29.3% HbA production in mobilized patient CD34+ cells, but long-term engraftment of only 2.3% in the bone marrow of mice [86]. More recently, 63% gene correction at the SCD mutation in human pluripotent cells [87], and 50% HbA expression in gene-corrected SCD CD34+ cells using HiFi Cas9 were reported [88], while only 2.5% engraftment of edited CD34+ cells were obtained in xenograft mouse bone marrow [89]. Though this list is far from exhaustive, taken together, it reveals the need for optimization of CRISPR/Cas9 editing techniques. Additionally, based on the donor chimerism data in allogeneic HSC transplantation, at least 20% gene correction is needed to reverse the sickle phenotype [90]. Nonetheless, despite imperfections with the CRISPR/Cas9 system, clinical trials are already underway and will be discussed in later sections.

### 3.5. Fetal Hemoglobin (HbF) Induction with Gene Editing

HbA is a tetramer protein composed of two α-globins and two β-globins. In SCD, the A-to-T conversion at position 20 in the β-globin gene (β^S^-globin) results in HbS production, which under hypoxic conditions, leads to polymerization and red blood cell sickling. In β-thalassemia, there is a deficiency in the production of β-globin chains and hence reduced levels of functional Hb causing anemia. In contrast, HbF contains γ-globin in place of β-globin. It is expressed until it is replaced by HbA at birth via a transcriptional switch [91,92]. HbF was discovered to have ameliorative effects in patients with both SCD and β-thalassemia. When patients are treated with hydroxyurea, an HbF inductor, vaso-occlusion can be reduced up to 50% [93,94]. This is not only due to the absence of the β^S^-globin mutant, but because HbF serves as an anti-sickling agent. This phenomenon was initially noted because children with SCD were asymptomatic until after infancy. It was also corroborated in asymptomatic SCD adults with abnormally elevated HbF levels. These individuals have a condition known as hereditary persistence of fetal hemoglobin (HPFH). The HPFH-related mutations are predominantly detected in specific regions including (1) the *BCL11A* gene, (2) the γ-globin promoter, (3) the *HBS1L*-*MYB* region, and (4) the δ- and β-globin gene locus [95]. HPFH mutations have become important targets in the development of gene-editing therapies in SCD due to consequential HbF induction [96].

Of the targetable HPFH regions, *BCL11A* has received a lot of attention. It plays a major role in Hb switching by directly binding to the γ-globin promoter, repressing HbF expression [97]. Using editing technologies such as CRISPR/Cas9, indels can be created at the *BCL11A* locus to disrupt its expression and increase HbF. This was successfully shown in human cells [98]. In SCD mouse models, it was shown that there was a direct link between *BCL11A* inactivation and amelioration of SCD symptoms [99]. Nonetheless, *BCL11A* plays varied roles in different hematopoietic lineages and is important in B-cell differentiation and HSC engraftment. Therefore, most studies have been geared toward targeting *BCL11A* erythroid-specific enhancers, the γ-globin promoter, and transcription factors (i.e., *GATA1*, *KLF1*, *FOG1*, *SOX6*, and the nucleosome remodeling deacetylase (NuRD) complex) [100,101], which have been shown to produce similar levels of HbF induction to disrupting *BCL11A* itself [96].

The *BCL11A* locus contains three DNase I hypersensitive sites (DHSs) named based on their distance from the transcriptional start site (+55, +58, +62) [102]. Of these, the DHS +58 is the most promising target for gene editing and includes the *GATA1* binding region. A study found that deletion of the DHS +58 in a human umbilical cord blood-derived erythroid progenitor-2 (HUDEP-2) cell line using CRISPR/Cas9 led to an increase in γ-globin from negligible to around 60% of the total β-like globins [100]. Additionally, it increased HbF expressing cells from around 5% to 40% [100]. Lesser HbF induction was found in disrupting the other sites (+55 and +62). A separate study using *BCL11A* enhancer-modified CD34+ cells resulted in a detectable increase in γ-globin, and did not negatively affect CD34+ cell proliferation, differentiation, and survival. Additionally, the edited cells could be expanded ex vivo and had a survival advantage due to benefits associated with γ-globin induction [103]. Engraftment of +58 CRISPR/Cas9-edited cells in mouse and rhesus models also demonstrated effective editing, therapeutic HbF induction levels, and low toxicity [104,105].

Other important targetable locations for HbF expression include the binding sites of *LRF*, *KLF1*, and *BCL11A* upstream of the γ-globin promoter. *LRF* is an HbF repressor that binds in the γ-globin promoter 200 bp upstream of the transcription start site and functions independently of *BCL11A* [106]. Using CRISPR/Cas9 to mimic HPFH mutations in the *LRF* binding site, γ-globin induction and correction of the sickling phenotype can be achieved in CD34+ cells [107]. *KLF1* also plays a role in HbF switching. It binds to the β-globin promoter and upregulates its expression. When mutated, HbF expression increases [108]. When knocked down via CRISPR-based editing in HUDEP-2 cells, γ-globin was upregulated 10-fold, *BCL11A* was downregulated 3-fold, and F-cells composed 20% of total cells. In *KLF1*-edited mobilized SCD CD34+ cells, HbF levels were between 40–60% of total Hb. Finally, disrupting the *BCL11A* binding site upstream of the γ-globin promoter via CRISPR/Cas9 editing increased HbF production in CD34+ cells by ≤40% with no deleterious effects on hematopoiesis [109]. These promising advances made in CRISPR/Cas9 editing to induce HbF production have led to the approval of several *BCL11A* targeted clinical trials in patients with both SCD and β-thalassemia.

### 3.6. Base-Editing in SCD

Despite the prevalence of gene editing, many techniques still rely on DSBs followed by HDR or NHEJ. DSB-reliant editing technologies are inefficient and can lead to high levels of insertions, deletions, translocations, and activation of the *P53* pathway [110,111]. Additionally, HDR is ineffective in quiescent cells. However, for diseases such as SCD (20A<T) or the β-thalassemia A>G variant that is caused by well-documented point mutations, a new technology known as ‘base editing’ provides a suitable non-DSB-reliant alternative. Additionally, the upstream region of γ-globin promoter which includes transcriptional factor binding sites, such as *BCL11A*, *LRF*, *KLF1*, and *GATA1* are also possible base-editing targets for HbF induction to treat SCD and β-thalassemia [112,113,114].

In 2016, the first ‘base editor (BE)’ was created, allowing for the conversion of a CG base pair to TA without using DSBs. This first-generation cytidine BE (CBE1) fuses nuclease-dead Cas9 (dCas9) with a cytosine deaminase [115]. The CBE opens the DNA creating an ‘R-loop’ that turns C to U and then relies on DNA mismatch repair systems to correct the complementary G to A. However, newer generations of BEs, such as BE3s, use rat variants of cytidine deaminases such as *APOBEC*, restore partial catalytic activity to dCas9 at the histidine residue, and attach DNA glycosylase inhibitors to the constructs as well, which prevent reversions of U to G. The D10A Cas9 variant ‘nickase’ allows Cas9 to nick the unedited strand and enhance mismatch repair editing. BE3s reported up to 37% conversion of C to T [115,116]. The newest and most effective generation of base editors (BE4 and BE4max) increases the editing efficiency of C:G to T:A by 50% and halves the frequency of undesired byproducts compared to BE3 [116]. SaBE4 editors and bacteriophage Mu Gamma (*Gam*) fusion to existing editors are also viable alternatives that increase efficiency and decrease indel formation [116].

CBE technologies have been successfully used to edit the +58 *BCL11A* erythroid-specific enhancer in human CD34+ cells. N57Q-BE3 (A3A), which has an engineered human *APOBEC3A* domain, achieved an 81.7% base conversion, which resulted in ~40% HbF induction in biallelic edited cells [117]. For β-thalassemia models, BE3 precisely edited the -28 (A>G) mutation in patient-derived fibroblasts and embryos with >23% editing efficiency [118]. A different BE3 model, eA3A-BE3, increased editing efficiency 40-fold compared to BE3 when targeting a point mutation in the β-thalassemia promoter region [119].

Nonetheless, CBEs only allow for CG to AT conversions. In 2017, the first adenine base editors (ABEs) were reported, which convert AT to GC. ABEs employ catalytically impaired Cas9 nickases fused to TadA, a deoxyadenosine deaminase enzyme. TadA was evolved from the *Escherichia coli* tRNA adenosine deaminase enzyme so that it could use single-strand DNA as a substrate. After ABE optimization, generation 7.10 averaged ~50% conversion rates in HEK293T cells with high product purity (99.9%) and low rates of indels (<0.1%) [120]. Recently, 8th generation ABEs were created as both NGG and non-NGG PAM variants with an overall higher on-target efficiency than ABE7.1 [121]. Although CBEs and ABEs cannot create transversion mutations, such as that needed to correct the SCD mutation 20A<T (Glu6Val), ABEs can replace the AT with a GC resulting in the benign G-Makassar phenotype. In this variant, individuals have normal hematological indices and no evidence of sickling. In CD34+ cells from SCD patients, ABE7.1 editing resulted in the conversion of HbS to HbG-Makassar at levels >80% leading to a reduction in HbS levels to less than 15% [122]. ABE8-NRCH created the non-pathogenic HbG-Makassar variant in SCD CD34+ cells with 80% HbS to HbG-Makassar conversion. After 16 weeks post-xenograft transplant, HbG-Makassar levels still accounted for 79% of β-globin protein in human erythroid cells, and sickling was reduced by 3-fold [123]. Additionally, ABEs have been employed to target the γ-globin promoter regions to increase γ-globin expression. While ABE7.1 only yielded the TA to CG mutations with ~30% efficiency in HEK293T cells [118], 8th generation ABEs produced much better results. In CD34+ cells in vitro, the γ-globin promoter could be edited in a dose-dependent manner. >80% editing was achieved in SCD CD34+ cells, which led to >60% γ-globin protein levels [124]. Base-edited human CD34+ cells led to long-term engraftment in immunodeficient mice (>90% human chimerism), maintained the base-editing variant (>90%), and displayed multi-lineage hematopoietic reconstitution. The edited erythroid cells also yielded >65% γ-globin levels, compared to unedited cells (<1.5%) [124].

Nonetheless, BEs still have several limitations including bystander cytosine deamination, PAM site restrictions, and the inability to catalyze all 12 possible base-to-base conversions. However, bystander cytosine deamination can be reduced by narrowing the editing frame [125,126]. Additionally, several ABE and CBE permutations have been generated with alternative PAM sites that increase the scope of this technology [127]. There have even been near PAM-less ABEs and CBEs created [128,129,130]. The T>A transversion mutation needed to treat SCD has yet to be attained via base editing. However, a new technology, known as ‘prime editors (PEs)’, can fill in where base editors fall short. Similar to base editing, it does not require DSBs but can generate all 12-point mutations. It can also install deletions of at least 80 nucleotides and insertions of at least 44 nucleotides with incredible precision and few off-target edits [53,131,132]. PEs fuse a dCas9 to an engineered reverse transcriptase with a prime editing guide RNA (pegRNA) [129]. Compared to BEs, PEs are more efficient at positions outside the center of the editing frame, but BEs are still more efficient within the frame. They have been successfully employed in generating and correcting the SCD mutation of the β-globin gene in HEK293T cells [129]. PEs have a lot of potentials, but they currently serve as a complementary technology to base editing.

### 3.7. Off-Target Effects with Gene Editing

Although site-specific endonucleases are used to achieve genome modification through DNA cleavage at the targeted locus, off-target effects may occur at genomic loci with high homology to the target sequence. The introduction of nucleases targeting the β-globin SCD mutation sometimes results in off-target activity at the δ-globin gene due to high sequence similarity. Analysis of ZFN-cleavage sites detected off-target activity in the δ-globin gene, instead of the targeted β-globin gene [130]. In CD34+ cells edited with ZFNs, intergenic deletion between on-target β-globin and off-target δ-globin sites was the most common rearrangement, followed by inversion of the intergenic fragment [131].

Similarly, the CRISPR/Cas9 system also suffers from undesired off-target effects, potentially leading to additional point mutations, deletions, insertions, or inversions. Therefore, high-fidelity versions of Cas9 have been developed to reduce DNA modifications at unintended genomic sites. A variant of the traditionally used *Streptococcus pyogenes* Cas9, Cas9-HF1 (high fidelity variant 1), can reduce most genome-wide off-target effects as compared to a wild-type SpCas9 [132]. Furthermore, HiFi Cas9 with a single point mutation (R691A) delivered via an AAV6 vector allowed for a 20-fold reduction in off-target effects as compared to a wild-type Cas9 [88]. Additionally, truncated guide RNAs of less than 20 nucleotides also retain Cas9 on-target editing and a ~5000-fold decrease in off-target effects [133].

Various methods exist for evaluating off-target effects, each with its advantages and disadvantages. Assays for detecting off-target activity include but are not limited to Genome-wide Unbiased Identification of DSBs Enabled by sequencing (GUIDE-seq), Circularization for In vitro Reporting of Cleavage Effects by sequencing (CIRCLE-seq), Breaks Labeling In Situ and Sequencing (BLISS), Discovery of In situ Cas Off-targets and Verification by sequencing (DISCOVER-seq), and various web-based prediction tools. In GUIDE-seq, double-stranded oligodeoxynucleotides are integrated at DSBs and are then mapped at the nucleotide level [134]. This method is unbiased, highly sensitive, and can detect off-target mutagenesis frequencies as low as 0.1%. However, GUIDE-seq is limited by chromatin accessibility [135]. In CIRCLE-seq, circularized genomic DNA with Cas9 cleavage sites is linearized and releases newly cleaved DNA ends [136]. Despite its sensitivity, further studies have yet to elucidate whether low-frequency Cas9-HF1-generated off-target mutations below the detection limit of GUIDE-seq are detectable by CIRCLE-seq. BLISS-seq offers a versatile, sensitive, and quantitative method for detecting endogenous and exogenous DSBs with low-input requirements [137]. Another method, DISCOVER-seq, leverages DNA repair factors recruited to DSB sites with applications in primary cells and in situ [138]. Although methods for off-target detection are advancing, developing reliable, highly sensitive, unbiased, and accurate assays remains a challenge in gene editing.

### 3.8. Clinical Trials of Gene-Editing Therapy in SCD and β-Thalassemia

While there have been ongoing lentiviral gene-addition trials since the early 2000s, gene-editing studies have only recently taken off (Table 2). Though most are trying to increase HbF expression through modification of the *BCL11A* enhancer, some are trying to correct the faulty β-globin gene ex vivo. The first clinical trial started in 2018 sponsored by Vertex Pharmaceuticals and is using a drug product called CTX001. It is composed of plerixafor-mobilized and CRISPR/Cas9-modified CD34+ cells and is being used to treat both TDT and SCD (NCT03655678, NCT03745287). CTX001 targets the *BCL11A* erythroid-specific enhancer [139]. In total, more than 40 patients have been enrolled since the study started. Fifteen TDT patients who ranged from 4–26 months post-treatment had total Hb ranging from 8.9–16.9 g/dL and HbF ranging from 67.3–99.6%. Seven SCD patients with data remained vaso-occlusion event-free 5–22 months post-gene-editing therapy with total Hb ranging from 11.0–15.9 g/dL and HbF from 39.6–49.6% at the last visit [140]. Other trials seeking to target *BCL11A* and HbF expression include Bioray’s Phase 1/2 trial in Shanghai for TDT (NCT04211480), Editas’ Ruby trial testing EDIT-301 for SCD (NCT04853576), Novartis Pharmaceuticals testing OTQ923 and HIX763 for SCD (NCT04443907), Sangamo Therapeutics testing ST-400 for TDT (NCT03432364), and Sanofi’s PRECIZN-1 trial testing SAR445136 (BIVV003) for SCD (NCT03653247). Bioray, Editas, and Novartis do not have any results reported. Interestingly, Editas’ EDIT-301 uses Cas12a instead of Cas9 (NCT04853576). Sangamo and Sanofi’s trials used ZFNs to modify the *BCL11A* gene. Sangamo’s ST-400 was infused in five patients. Though HbF levels peaked around 23.5±11.4%, these levels were not sustained, precluding patients from reaching transfusion independence. This is presumably due to low levels of edited long-term progenitors in the infused drug product [141]. In contrast, the Sanofi’s PRECIZN-1 study infused four patients with SCD with greater success. As of June 2021, 26 weeks post-infusion revealed stabilization of Hb (9–10 g/dL), HbF (14–39%), and %F-cells (49–94%) in all four subjects and no recurrence of previous vaso-occlusion events. No adverse events and severe adverse events were reported [142]. Three newer studies are attempting to correct the disease mutation in the β-globin gene in patients with TDT and SCD. Bioray Laboratories has a similar study to its γ-globin study where it uses β-globin-targeted guide RNA in TDT patients (NCT04205435). New Graphite Bio enrolled its first patient in November 2021 to its CEDAR study to test GPH101, a drug product composed of CRISPR/Cas9-edited CD34+ cells (NCT04819841) [143]. Finally, the University of California, San Francisco will test its Drug Product of CRISPR/Cas9-edited CD34+ cells from patients with SCD (NCT04774536).

## 4. In Vivo Gene Editing for SCD

In vivo gene editing has the potential to circumvent many challenges of ex vivo editing, which include cytotoxic myeloablation as well as the cost and complexity of ex vivo cell manipulation. Furthermore, the potential simplicity of in vivo approaches could allow for this cure to extend to populations in developing areas such as Sub-Saharan Africa where SCD is endemic. Successful in vivo gene-editing therapy requires precise delivery of gene-editing machinery to target tissues followed by a constitutive expression. Challenges to in vivo editing that need to be addressed include maximizing delivery to target cells, minimizing delivery to nontarget cells, reducing immunogenicity, and achieving robust expression.

Numerous methods have been developed for the delivery of genetic material or gene-editing cargos to cells in vitro [145]. As mentioned above, they are divided into two categories: viral and non-viral delivery systems. In the viral delivery system, gene-editing cargos are packaged into viral vectors (i.e., AAVs, Ads, and lentiviruses) [146]. The non-viral delivery system employs physical and chemical methods to deliver genetic material and editing machinery to cells. Physical delivery methods are commonly used for ex vivo delivery, such as electroporation and microinjection [84]. Chemical methods are generally mediated by nanoparticles including lipid, polymer, and gold, which are internalized via endocytosis or macro-pinocytosis [147].

For in vivo gene editing, CRISPR/Cas9 cargos can be delivered systemically or locally. During systemic delivery, CRISPR/Cas9 cargos are introduced into the body via intravenous injection, distributed by the circulatory system, extravasate from the blood vessels, migrate into the interstitial space, and enter target cells. Local delivery, characterized by direct injection of editing cargo into the interstitial space, minimizes dissemination into off-target tissue. However, it leads to heterogeneous distribution in the target tissue. Efficient in vivo delivery is trickier than ex vivo. Additionally, some techniques used on animals are unfit for humans. Despite these challenges, progress has been made in this field [148]. Below, we briefly describe methods commonly used in vivo delivery of gene-addition/editing tools and their relative efficacy.

### 4.1. In Vivo Delivery with AAV Vectors

AAV vectors have successfully been used to deliver CRISPR/Cas9 cargos in vivo in several animal models through mutant gene correction via HDR [149,150,151], transgene insertion [152,153], induction of exon skipping and reframing [154], and gene deletion or disruption [155,156,157]. The AAV capsid can be engineered to increase its tissue tropism. This means a lower vector dosage is necessary to achieve therapeutic levels, reducing the side effects. Recently, an AAV vector was modified by inserting a random peptide into the capsid, resulting in 10× more efficient targeting of skeletal muscle tissue [152]. It was successfully able to deliver the CRISPR/Cas9 cargos to mice and non-human primate muscle cells; and in the mouse model, it was able to repair dysfunctional copies of the dystrophin gene.

Although AAV vectors have great potential for in vivo gene therapy, they have several limitations which include small packaging capacity, immunogenicity, liver toxicity, prolonged Cas9/sgRNA expression, and possible integration of the viral genome. The most common SpCas9 is derived from *Streptococcus pyogenes*, which has a cDNA size of 4.1kb. The AAV can only hold about 4.7kb, placing SpCas9 near the packaging limit. Therefore, sgRNA and donor DNA should be packaged into a separate AAV vector to circumvent its limited capacity, which comes at the expense of editing efficiency [158,159]. Alternatively, smaller Cas9 orthologs can be used such as SaCas9 (3.2 kb) from *Staphylococcus aureus* [160], CjCas9 (2.95 kb) from *Campylobacter jejuni* [161], Cas12a (3.3 kb) from *Lachnospiraceae* or *Acidaminococcus* [162], Cas12b from the mutant *Bacillus hisashii* [163], and the CasX (Cas12e; 2.9 kb) from groundwater bacteria. These orthologs have robust editing activity, comparable to SpCas9.

Immunity against the AAV capsid due to its non-enveloped protein shell is another major limitation of this vector: 35–80% of humans have pre-existing neutralizing antibodies against AAVs, which block their entry into target cells [164,165]. Additionally, AAV vectors cannot be repeatedly administered due to antibody production against the AAV capsid, unless a different serotype is used. Nonetheless, engineered AAV capsids (chimeric AAV capsids or altering the antigen site) are partially able to escape humoral immunity [166]. However, vector modification may negatively impact its infectivity or tissue tropism [167]. Administration of the high-doses AAV vector can overwhelm the inhibitory effect of pre-existing neutralizing antibodies, but large doses also increase toxicity [168].

While AAV vectors typically exist in host cells as episomes, a high incidence of random integration into host genomes has been reported [169,170]. When mice were treated with AAV vectors encoding Cas9, on-target and off-target insertion throughout the chromosome was detected. This can lead to genotoxicity such as endogenous gene disruption and oncogene activation [170]. Constitutive Cas9 expression due to prolonged AAV vector transgene expression of up to 10 years is also a concern as it can lead to off-target mutations [171]. However, a self-cleaving AAV system has been developed to reduce Cas9 duration in a mouse model [172].

High doses of AAV vectors also result in liver toxicity. In the non-human primate model, the animals developed severe hepatotoxicity and morbidity within 4–5 days of intravenous injection using a high dose of the AAV vector [168]. A high copy number (> 1000 v.g.) of the vector was observed in the liver of dead animals, which was thought to be related to liver toxicity. Liver toxicity was also observed in three patients who passed away within 3–4 weeks after intravenous administration of a high dose of AT-132 in an AAV vector gene therapy trial in X-linked myotubular myopathy [173]. Some patients also developed progressive cholestatic hepatitis and subsequent decompensated liver failure. Taken together, all these limitations restrict the utility of AAV vectors for in vivo gene-editing therapy in clinics.

### 4.2. In Vivo Delivery with Ad Vectors

Ad vectors were first used in vivo to deliver the *A1AT* gene in rat hepatocytes and lung tissues in the early 1990s [174,175]. Due to preliminary successes, clinical trials using Ad vector-based in vivo gene delivery to treat monogenic diseases were launched. Two Ad in vivo gene-addition trials were performed, including *CFTR* gene addition to patients’ nasal tissues in cystic fibrosis, and *VEGF* gene addition to patients’ cardiac muscles in coronary artery disease [176,177]. Ad vectors are strongly immunogenic, which causes poor delivery and low transgene expression [178]. In a gene therapy trial for ornithine transcarbamylase deficiency, a hepatic artery injection of an Ad vector resulted in patient death in 1999 due to a cytokine storm triggered by the capsid protein. Since then, the use of Ad vectors has declined in gene therapy [179]. Even modified Ad vectors triggered immunoresponses, resulting in severe side effects [180]. Furthermore, the high prevalence of pre-existing antibodies against Ad capsids limits the clinical application of Ad vectors [181,182].

In vivo gene delivery was successfully demonstrated with a CD46-targeted Ad vector (HDAd5/35++ vector) encoding a transposon-based integration system in mice and non-human primates [183,184]. The HDAd5/35++ vector can also systemically deliver gene-editing tools and site-specific integration at the *AAVS1* gene (a safe harbor site, allowing for the insertion without oncogenesis) in a mouse model [185]. While this approach was specifically designed for in vivo gene delivery to HSCs, it has a low editing efficiency and poor tissue specificity. Serial drug selections are required for the detection of gene editing in vivo. In addition, Ad vectors were recently used for COVID-19 vaccination (AstraZeneca and Johnson & Johnson) [186,187]. The vaccine, though effective, is limited to a one-time injection due to neutralizing antibody production.

### 4.3. In Vivo Delivery with LVs

LVs have a large genome capacity of about 7–10 kb [188] and have been used in many clinical trials for ex vivo gene-addition therapy to treat monogenetic diseases. Nonetheless, insertional mutagenesis is still a concern when using them. Systemic injection of LVs in mice induced immunoreaction to the transgene products rather than the LVs themselves, suggesting that LVs are less immunogenic [189]. IDLVs have been developed to prevent lentiviral integration into the targeted cell genome with efficient transduction [190]. IDLVs can deliver ZFN as well as CRISPR/Cas9 cargo to a broad range of dividing and quiescent tissues [191,192]. However, only a few studies of in vivo gene editing using LV delivery of CRISPR/Cas9 cargos have been reported in model animals. One study demonstrated that intratracheal inoculation or direct intrapulmonary injection of an LV with CRISPR editing machinery was able to induce *Eml4-Alk* gene rearrangement in lung cells in mice [193]. Subretinal injection of IDLVs was also shown to selectively ablate the *Vegfa* gene in mice, resulting in the disruption of *Vegfa* with an in vivo indel formation efficacy of up to 84% [194]. These studies indicate that IDLV-based CRISPR/Cas9 editing is potentially a feasible and simple method for in vivo gene therapy.

### 4.4. In vivo Delivery with Virus-Like Particles (VLPs)

VLPs are nanoparticles derived from the self-assembly of viral structural proteins, but lack the viral genome, making them replication incompetent, similar to IDLVs [195,196]. LV-based VLPs can carry up to a ~8kb cargo, which is embedded inside the capsid itself. They are also transiently expressed, have minimal risk of integration, and have envelopes that can be extensively modified for specificity [197]. Several VLP systems have been developed, including the Cas9P LV that pre-packages the Cas9 protein in lentiviral particles [198], the VEsiCas system that passively incorporates SpCas9 [199], the NanoBlades system that fuses SpCas9 with retroviral Gag [200], or the Gesicle system that uses dimerization-based incorporation of SpCas9 [201]. Though VLPs have been primarily used in vaccines clinically, there is only one ongoing Phase 1/2 VLP-based CRISPR/Cas9 RNA delivery study using a corneal injection for herpetic stromal keratitis in Shanghai (NCT04560790).

### 4.5. In Vivo Delivery with LNPs

Since the creation of the LNP in 1990, it has become a critical tool in the world of gene editing, pharmaceuticals, and vaccinations [202]. LNPs provide good alternatives to viral delivery methods, along with transient CRISPR expression, a few payload restrictions, broad tissue tropism (i.e., muscle, brain, liver, and lungs), no preexisting immunity, and low immunogenicity. Due to these benefits, they have been employed in several successful in vivo studies, including small interfering RNA (siRNA) therapy as well as COVID-19 mRNA vaccination (Moderna or Pfizer) [203,204].

Due to the negative charge of DNA and RNA, they can be encapsulated inside positively charged lipids, including DOTAP (1,2-Dioleoyl-3-trimethylammonium) [205]. This ensures their protection from nuclease degradation and immunological response, but the permanent cationic lipids are toxic [206]. Therefore, a new generation of cationic lipids was developed to be positively charged at low pH but neutral in physiological pHs. For example, MC-3 (dilinoleylmethyl-4-dimethylaminobutyrate), which was used for the first FDA-approved siRNA drug with LNPs [207,208]. Generally, LNPs for mRNA delivery are composed of new-generation cationic lipids as well as phospholipids, cholesterol, and/or polyethylene glycol (PEG) lipids to improve the stability and delivery efficiency [208,209]. Additionally, the zwitterionic lipid can escape from the endosome, resulting in the efficient delivery to the liver, kidneys, and lungs of mice [210]. A later study developed a single administration, biodegradable, LNP delivery system that achieved robust editing of the transthyretin (*Ttr*) gene in the liver of a mouse. It led to a >97% reduction in serum protein levels for 12 months [211].

While LNPs have been used in other clinical settings, the first clinical trial for LNP-based CRISPR delivery was launched in 2020 to treat patients with hereditary transthyretin amyloidosis (ATTR) (NCT04601051). ATTR, caused by mutations in the *TTR* gene, is characterized by amyloid plaque buildup in the body’s organs and tissues and affects the peripheral nervous system. In the study, systemic administration with LNP-based CRISPR/Cas9 delivery to the liver (drug product: NTLA-2001) was able to safely knock out the mutant *TTR* gene in a dose-dependent manner [212]. While this study has produced promising results, optimization is still required and therefore, it has yet to be adapted and applied to SCD.

### 4.6. In Vivo Delivery with Polymer Nanoparticles (PNPs)

PNPs are another non-viral delivery system that has been successfully employed in genetic editing due to their ability to encapsulate, stabilize, and protect genetic cargo from degradation and immunoreaction. Nucleic acid-PNP complexes can be generated via non-covalent encapsulation or covalent derivatization. Compared to LNPs, PNPs are structurally more dynamic as they can be engineered to take on a greater array of chemical forms, potentially allowing for greater tissue specificity.

The most common PNP is polyethyleneimine (PEI). It can condense and encapsulate nucleic acids efficiently due to its high charge density. However, PEIs are toxic due to their strong positive charges. Therefore, PEIs are mainly used for plasmid transfection to culture cells in vitro. Lower weight PEIs (i.e., PEI 25K) or moieties such as PEG can be added to decrease immunogenicity, increase systemic circulation time, and increase editing efficiency. Another popular PNP is the multistage delivery nanoparticle system (MDNP), composed of a cationic core and negatively charged shell [213]. The negativity of its shell allows it to remain in circulation longer while its positive interior allows strong binding and stabilization of the nucleic acids it carries.

Several studies highlight the success of PNP-based CRISPR/Cas9 delivery in vitro and in vivo. Polyethylene glycol-b-poly lactic-glycolic acid (PEG-b-PLGA) successfully delivered CRISPR machinery to macrophages [214]. A study using PEGylated cationic nanoparticles (named P-HNPs) achieved 47.3% gene editing in cultured cells and 35% gene editing in a mouse model in vivo, inhibiting tumor growth [215]. An MDNP carrying dCas9-miR-524 was systemically delivered to mice, leading to effective upregulation of miR-524 and tumor growth suppression [213]. Finally, a poly (lactic-co-glycolic acid) (PLGA) nanoparticle was used for systemic delivery of triplex-forming peptide nucleic acids (a recombination tool) and donor DNA, allowing for *CCR5* gene editing in human blood mononuclear cells in immunodeficient mice [216]. While this technology is advancing, there are currently no human clinical trials using it for CRISPR delivery.

### 4.7. In Vivo Delivery with AuNPs

AuNPs are composed of a gold core (a few-to-several hundred nanometers in size), which can be coated with synthetic or biological compounds. These coated compounds can be covalently or non-covalently conjugated to the therapeutic molecules (i.e., DNA, RNA, and proteins) [217]. The non-covalently conjugated drugs can be simply released in the target cells, while a specific cleavage method is required for the covalent conjugated AuNPs, including glutathione-mediated release [218] and thermally triggered release [219]. AuNPs are advantageous because they can be easily synthesized to form different shapes and sizes, can take on a plethora of moieties to increase specificity and function, have a positive charge (binding to DNA and RNA), and are not toxic. Unlike LNPs and PNPs, the CRISPR cargos are bound on the surface of AuNPs, possibly inducing a greater immunoreaction in patients. However, the binding site of cargos is hidden by AuNPs from immune cells [220]. In addition, AuNPs are thought to have anti-inflammatory activity, since gold is utilized to reduce inflammation in rheumatoid arthritis [221]. Therefore, the immunogenicity might be milder in AuNP-based in vivo delivery, compared with the direct injection of drugs.

Though AuNPs have yet to be used for CRISPR delivery in humans, several studies report effective in vitro and in vivo editing. An AuNP-based CRISPR delivery achieved 90% delivery efficiency and up to 30% editing efficiency in a HeLa cell line [222]. A layer-by-layer AuNP-based CRISPR delivery using the Cas12a nuclease allowed for 13.4% HDR at the *CCR5* gene and 8.8% HDR at the γ-globin promoter in CD34+ cells in vitro [223]. The editing efficiencies were comparable to electroporation-mediated delivery, suggesting that AuNP-based CRISPR delivery can be used for gene editing in CD34+ cells. In a melanoma mouse model, lipid-coated AuNPs knocked out the *Plk-1* gene in vivo, following intertumoral injection and laser-mediated thermal activation [224]. An intramuscular injection of CRISPR-AuNPs resulted in a 5.4% gene correction in vivo in Duchenne muscular dystrophy mice [225]. AuNPs have great potential as a future in vivo delivery system for SCD. Nonetheless, the data on off-target delivery, immunogenicity, and long-term effects remain limited.

## 5. Conclusions

Although recent HSC-targeted gene-addition and gene-editing therapies for SCD have produced encouraging results, these therapies still face many of the challenges discussed above. While LV gene addition has proven to be an effective and relatively safe therapy, gene editing allows to fix the SCD mutation without the risk of LV integration. Higher efficacy, lower cost, and minimal safety concerns could be achieved through optimization of conditioning, CD34+ cell collection, therapeutic vector design, and gene delivery methods. Simplification and standardization of this process are desired as well. Long-term follow-up of the ex vivo trials discussed above will also be informative in assessing the future role of gene editing and addition in the treatment of disease. Though requiring further development, in vivo treatment holds promise and its attainment is the primary objective of many researchers in the field as they believe it could make gene therapy the global standard of care for patients with SCD.

## Figures and Tables

**Figure 1 cells-11-01843-f001:**
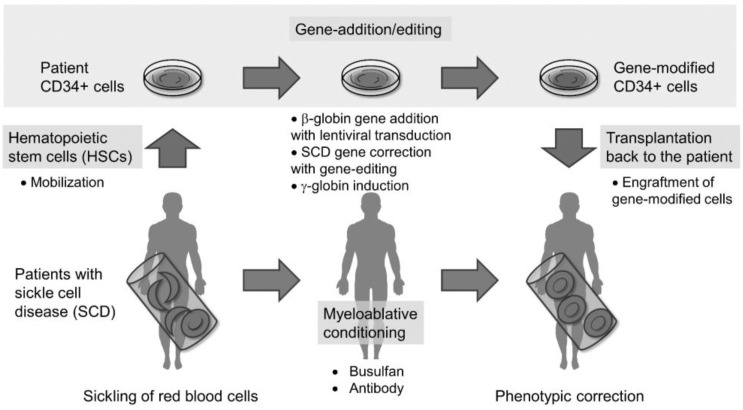
Overview of hematopoietic stem cell (HSC)-targeted gene therapy in sickle cell disease (SCD). A schema of discussion points in autologous HSC therapy with gene addition/editing in SCD.

**Figure 2 cells-11-01843-f002:**
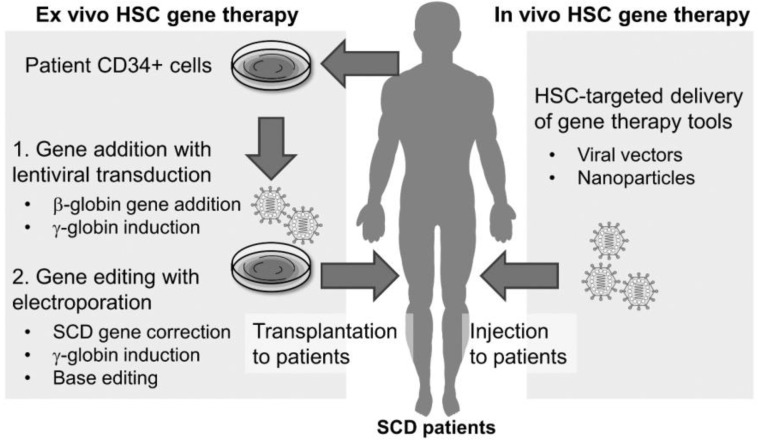
Comparison between ex vivo and in vivo HSC-targeted gene therapies in SCD.

**Table 1 cells-11-01843-t001:** A summary of lentiviral HSC gene-addition therapy trials in SCD and transfusion-dependent β-thalassemia (TDT).

Trial	Phase	Year	Study Drug	Target Gene	HSC Source	Conditioning	Sponsor	Location	Reference
Sickle cell disease									
NCT04628585	N/A	2020	BB305	β^T87Q^-globin	N/A	N/A	bluebird bio	USA, France	[62]
NCT04293185	3	2020	BB305	β^T87Q^-globin	Plerixafor mobilization	Myeloablative busulfan	bluebird bio	USA	N/A
NCT04091737	1	2019	CSL200	γ^G16D^-globin, shRNA-HPRT	Plerixafor mobilization	RIC melphalan	CSL Behring	USA	N/A
NCT03964792	1/2	2019	GLOBE1	β^AS3^-globin	Plerixafor mobilization	Myeloablative busulfan	APHP	France	[68,69]
NCT03282656	1	2018	BCH-BB694	shmiR-BCL11A	BM	Myeloablative busulfan	Boston Children’s Hospital	USA	[74]
NCT02247843	1/2	2014	βAS3-FB	β^AS3^-globin	Plerixafor mobilization	Myeloablative busulfan	University of California, Los Angeles	USA	N/A
NCT02186418	1/2	2014	ARU-1801	γ-globin	BM/Plerixafor mobilization	RIC melphalan	Aruvant Sciences	USA, Canada, Jamaica	[72]
NCT02140554	1/2	2014	BB305	β^T87Q^-globin	BM (Group A/B), Plerixafor mobilization (Group C)	Myeloablative busulfan	bluebird bio	USA	[38]
NCT02151526	1/2	2013	BB305	β^T87Q^-globin	BM	Myeloablative busulfan	bluebird bio	France	[62,68]
β-Thalassemia									
NCT05015920	1	2021	BD211	β^T87Q^-globin	BM	Myeloablative busulfan	Shanghai Bdgene	China	N/A
NCT04592458	1	2020	LentiHBBT87Q	β-globin	BM	N/A	Shenzhen Children’s Hospital, BGI-Research	China	N/A
NCT03207009	3	2017	BB305	β^T87Q^-globin	G-CSF and plerixafor mobilization	Myeloablative busulfan	bluebird bio	USA, France, Germany, Greece, Italy, UK	[65]
NCT02906202	3	2016	BB305	β^T87Q^-globin	G-CSF and plerixafor mobilization	Myeloablative busulfan	bluebird bio	USA, France, Germany, Italy, Thailand, UK	[65]
NCT02453477	1/2	2015	GLOBE	β-globin	G-CSF and plerixafor mobilization	Myeloablative treosulfan/thiotepa	TIGET	Italy	[68]
NCT01745120	1/2	2013	BB305	β^T87Q^-globin	BM	Myeloablative busulfan	bluebird bio	USA, Australia, Thailand	[64]
NCT02151526	1/2	2013	BB305	β^T87Q^-globin	BM	Myeloablative busulfan	bluebird bio	France	[64]
NCT02633943	N/A	2013	BB305		BM/G-CSF and plerixafor mobilization	Myeloablative busulfan	bluebird bio	USA, Australia, France, Germany, Italy, Thailand, UK	[62]
NCT01639690	1	2012	TNS9.3.55	β-globin	G-CSF mobilization	RIC busulfan	Memorial Sloan Kettering	USA	[45,67]
LG001	1/2	2007			BM	Myeloablative busulfan	bluebird bio	France	[50,60]

BM: bone marrow, RIC: reduced-intensity conditioning, N/A: not applicable.

**Table 2 cells-11-01843-t002:** A summary of HSC gene-editing therapy trials in SCD and TDT.

Trial	Phase	Year	Study Drug	Target Gene	HSC Source	Conditioning	Sponsor	Location	Reference
Sickle cell disease									
NCT04774536	1/2	2021	CRISPR/Cas9: CRISPR_SCD001	β-globin	BM	Myeloablative Busulfan	University of California, San Francisco	USA	N/A
NCT04853576	1/2	2021	CRISPR/Cas12: EDIT-301	*BCL11A* ESE	Plerixafor mobilization	Myeloablative busulfan	Editas Medicine	USA	N/A
NCT04819841	1/2	2021	CRISPR/Cas9: GPH101	β-globin	BM	Myeloablative busulfan	Graphite Bio	USA	N/A
NCT05145062	N/A	2021	ZFN: BIVV003	*BCL11A* ESE	Plerixafor mobilization	Myeloablative busulfan	Bioverativ	USA	[142]
NCT04443907	1/2	2020	CRISPR/Cas9: OTQ923 / HIX763	*BCL11A* ESE	N/A	N/A	Novartis Pharmaceuticals	USA	N/A
NCT03653247	1/2	2019	ZFN: BIVV003	*BCL11A* ESE	Plerixafor mobilization	Myeloablative busulfan	Bioverativ	USA	[142]
NCT04208529	N/A	2019	CRISPR/Cas9: CTX001	*BCL11A* ESE	BM	Myeloablative busulfan	Vertex Pharmaceuticals	USA	[144]
NCT03745287	2/3	2018	CRISPR/Cas9: CTX001	*BCL11A* ESE	BM	Myeloablative busulfan	Vertex Pharmaceuticals	USA	[144]
β-Thalassemia									
NCT04205435	1/2	2021	CRISPR/Cas9	β-globin	N/A	N/A	Biorary Laboratories	China	N/A
NCT04208529	N/A	2021	CRISPR/Cas9: CTX001	*BCL11A* ESE	BM	Myeloablative busulfan	Vertex Pharmaceuticals	USA	[144]
NCT04211480	1/2	2020	CRISPR/Cas9	*BCL11A* ESE	N/A	N/A	Biorary Laboratories	China	N/A
NCT03655678	2/3	2018	CRISPR/Cas9: CTX001	*BCL11A* ESE	BM	Myeloablative busulfan	Vertex Pharmaceuticals	USA, Canada, Germany, Italy, UK	[144]
NCT03432364	1/2	2018	ZFN: ST-400	*BCL11A* ESE	G-CSF & plerixafor mobilization	Myeloablative busulfan	Sangamo Therapeutics	USA	[141]

ESE: erythroid-specific enhancer, BM: bone marrow, RIC: reduced-intensity conditioning, N/A: not applicable.

## Data Availability

Not applicable.

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
