# Peer review of "Hematopoietic Stem Cell Gene-Addition/Editing Therapy in Sickle Cell Disease"

_cells, 2022, doi:10.3390/cells11111843_

Round 1
Reviewer 1 Report
This review manuscript describing the cure method for sickle cell disease mainly using HSC gene-addition/editing is very interesting and contains various information regarding to these methods per se, and their clinical trials. The authors described gene-addition/editing in detail, which will be understandable for all readers including unexperienced researchers. Thus the manuscript is well written, I have a few comments.
Line 139, efficient gene marking in CD34+ cells → What gene? Also in line 173, gene marking levels, does this gene mean added gene?
Line 194, is SIN an abbreviation?
Lines 232-235, I could not understand what the authors would like to mention. How robust engraftment of HSCs with high-level gene marking is important to maintain polyclonal hematopoiesis?
Reviewer 2 Report
The manuscript of Germino-Watnick et al. with the title "Hematopoietic stem cell gene-addition/editing therapy in sickle cell disease" presents an informative study that aims to describe the prospects of HSC gene therapy for SCD and β-thalassemia. This study is descriptive, detailed, and provides key details in the field of gene therapy. However, there are several minor issues that need to be addressed:
· The authors are advised to provide additional information with regard to the reason for which they studied β-thalassemia and what it offers in terms of novelty.
· Line 48: “Targeting γ-globin expression through its regulatory elements such as BCL11A, is another approach for HSC editing”. It is suggested that the authors provide additional information on γ-globin regulatory elements and how they are involved in HSC editing.
· “However, insertional mutagenesis remains a risk. Additionally, achieving long-term engraftment of gene-modified cells post-transplant can be challenging as culture conditions must adhere to certain guidelines”. The authors should mention how these limitations could be overcome.
· The authors should clarify whether the combination of plerixafor/G-CSF is indicated in HSD patients or not.
· In the third and fourth part, the authors describe the gene-editing techniques in detail. However, the authors need to highlight the technique with the most advantages for HSD.
· Due to the text extensive length, the addition of figures, especially in part 3, would render the text more comprehensible.
· Additional commentary could be added in the Conclusions, regarding the importance of gene editing in SCD and their potential utilization in standard clinical practice.
· Proofreading of the manuscript is needed, since some syntax and grammar errors exist.
Round 2
Reviewer 1 Report
I do not have further comments.